# Many-Shot In-Context Learning

Rishabh Agarwal[*], Avi Singh[*], Lei M. Zhang[†], Bernd Bohnet[†], Luis Rosias[†], Stephanie Chan[†], Biao Zhang[†], Ankesh Anand , Zaheer Abbas , Azade Nova , John D. Co-Reyes , Eric Chu , Feryal Behbahani , Aleksandra Faust  and Hugo Larochelle

[*]Contributed equally, [†]Key contribution

Large language models (LLMs) excel at few-shot in-context learning (ICL) – learning from a few input-output examples ("shots") provided in context at inference, without any weight updates. Newly expanded context windows allow us to investigate ICL with hundreds or thousands of examples – the many-shot regime. Going from few-shot to many-shot, we observe significant performance gains across a wide variety of generative and discriminative tasks. While promising, many-shot ICL can be bottlenecked by the available amount of human-generated outputs. To mitigate this limitation, we explore two settings: (1) "Reinforced ICL" that uses model-generated chain-of-thought rationales in place of human rationales, and (2) "Unsupervised ICL" where we remove rationales altogether, and prompt the model only with domain-specific inputs. We find that both Reinforced and Unsupervised ICL can be effective in the many-shot regime, particularly on complex reasoning tasks. Furthermore, we demonstrate that, unlike few-shot learning, many-shot learning is effective at overriding pretraining biases, can learn high-dimensional functions with numerical inputs, and performs comparably to fine-tuning. Finally, we reveal the limitations of next-token prediction loss as an indicator of ICL performance.

## 1. Introduction

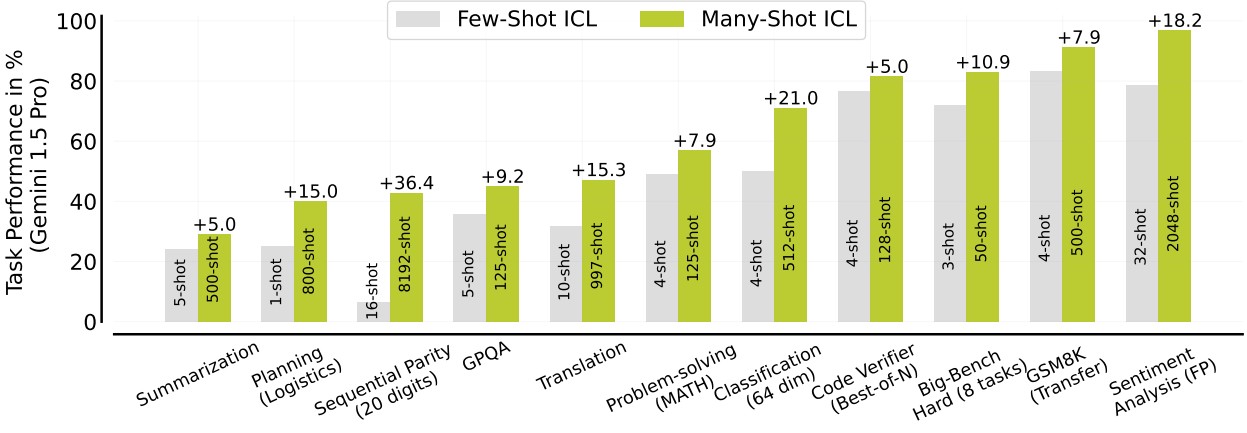

Figure 1 | **Many-shot vs Few-Shot In-Context Learning** (ICL) across several tasks. Many-shot ICL consistently outperforms few-shot ICL, particularly on difficult non-natural language tasks. Optimal number of shots for many-shot ICL are shown inside the bar for each task. For few-shot ICL, we either use typical number of shots used on a benchmark, for example, 4-shot for MATH, or the longest prompt among the ones we tested with less than the GPT-3 context length of 2048 tokens. Reasoning-oriented tasks, namely MATH, GSM8K, BBH, and GPQA use chain-of-thought rationales. For translation, we report performance on English to Bemba, summarization uses XLSum, MATH corresponds to the MATH500 test set, and sentiment analysis results are reported with semantically-unrelated labels. See §2, §3, and §4 for more details.

A limiting factor for *in-context learning* (ICL) in LLMs is the context window, restricting prior research to the *few-shot* ICL regime. *Many-shot* learning – ICL with a large number of shots, for example, hundreds or thousands – allows for better task specification, can reduce the need for fine-tuning, and potentially make LLMs more versatile and adaptable. Exploring many-shot ICL is now

arXiv:2404.11018v2 [cs.LG] 22 May 2024

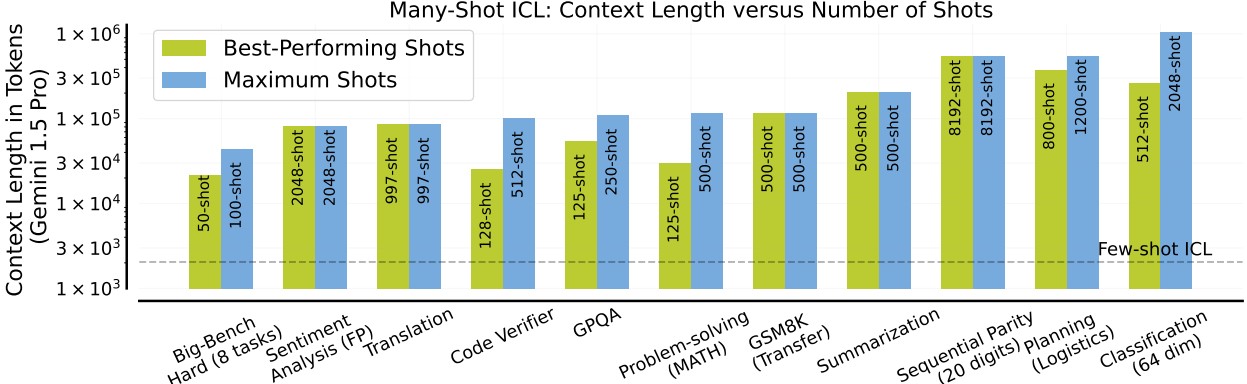

Figure 2 | **Context Length** for best-performing and the maximum number of shots tested for each task. The horizontal dashed line shows the context length of GPT-3 (2048 tokens), which is representative of typical few-shot prompts tested in the LLM literature. For several tasks, we observed the best-performing shots correspond to the maximum number of shots we tested, which was often limited by the number of available examples for in-context learning. On some tasks (e.g., code verifier, planning), we did observe slight performance deterioration beyond a certain number of shots.

feasible, given the recent increase in context windows of publicly available LLMs by at least 100×: from only a few thousand tokens in GPT-3 [8] and Llama 2 [57] to 1M tokens in Gemini 1.5 Pro [16].

In this paper, we investigate how scaling the number of shots affects ICL performance on a wide variety of tasks (§2): problem solving using MATH [23] and GSM8K [10], question-answering [GPQA, 52], summarization using XSum [43] and XLSum [20], algorithmic reasoning [BBH, 56], reward modeling [Code Verifier, 24], low-resource machine translation [FLORES, 18], planning [Logistics, 54], and sentiment analysis [FP, 40]. Compared to few-shot ICL, many-shot learning performs significant better across these tasks, using several hundreds or thousands of shots (Figure 1). Furthermore, maximum performance is often achieved only once the number of shots reaches up to *hundreds of thousands* of tokens (Figure 2). Concurrent to our work, recent works explore many-shot ICL to jailbreak LLMs [2] (up to 256 shots) and tackle NLP classification tasks [6] (up to 80K tokens). In our work, we focus on a much wider range of tasks, use a lot more examples (up to 8192 shots), and much longer context lengths (up to 1M tokens). See §5 for a detailed discussion of related work.

While many-shot ICL holds significant promise, it can be constrained by the need for high-quality, human-generated outputs. To overcome this, we introduce *reinforced* ICL and *unsupervised* ICL (§3). Inspired by the efficacy of model-generated solutions for fine-tuning [55], Reinforced ICL involves replacing human-written rationales with model-generated ones, filtered via answer correctness, for in-context learning. Inspired by task-recognition view of ICL [66], we also introduce Unsupervised ICL where we prompt the model with only problems instead of problem-solution pairs. On problem-solving tasks such as MATH, GPQA and Big-Bench Hard, we find that both reinforced and unsupervised ICL with many-shots can be more effective than few-shot ICL with human-generated rationales, with reinforced ICL being more broadly effective.

Finally, we empirically study how the learning dynamics of in-context learning changes from few-shot to the many-shot regime (§4). We find that with sufficient examples, ICL can overcome pre-training biases, perform comparably to full fine-tuning, and solve high-dimensional prediction tasks with numerical inputs, namely sequential parity prediction and linear classification. This suggests the potential of many-shot ICL to adapt to unseen tasks and domains that might be misaligned with an LLM's training data. Surprisingly, the order of examples can influence many-shot performance (§4.3) Finally, we demonstrate that long-context scaling laws [2, 68, 27] based on next-token prediction loss may not reliably predict ICL performance on problem-solving and reasoning tasks.

Our key contributions are as follows:

- **Scaling ICL** (§2): We systematically evaluate ICL performance at different scales of in-context examples for a wide range of tasks with Gemini 1.5 Pro. Our results indicate large performance jumps when transitioning from few-shot to many-shot regime.

- **Reinforced and Unsupervised ICL** (§3): We find that using model-generated rationales or only problems can reduce the dependence of many-shot ICL on human-generated data.

- **Analysing ICL** (§4): We show that many-shot ICL can overcome pre-training biases, perform comparably to fine-tuning, and learn non-NLP prediction tasks, where few-shot ICL struggles. We also reveal that next-token prediction loss may not be a good predictor of ICL performance.

## 2. Scaling In-Context Learning

During in-context learning (ICL), the LLM receives a prompt containing a set of input-output examples, also called *shots*, that illustrate the desired task. At the end of the prompt, we append a test input and allow the LM to make a prediction just by conditioning on the prompt and predicting the next tokens auto-regressively. Recent increase in context windows of LLMs allow using many more shots for ICL than typically used. Exposure to many more shots can lead to better generalization, handle more complex problems than what is possible with few-shot ICL, make fine-tuning less essential, and greater control over model outputs, potentially reducing biases stemming from pre-training.

**Evaluation**   We evaluate the many-shot performance of Gemini 1.5 Pro[1] [16] model with 1 million token context length, the largest publicly available so far. Unless specified otherwise, we use greedy decoding. For reliable results, we randomly sample in-context examples for each $K$-shot prompt multiple times using different random seeds and report average performance, along with some visualization for performance on individual seeds. To ensure that using more shots provides additional information, any $K$-shot prompt in our setup includes all in-context examples from prompts with less than $K$ examples. To reduce the inference cost, we use *KV caching* [49]. Next, we study many-shot ICL on typical LLM use-cases (also see §2.4).

### 2.1. Machine Translation

We consider translation from English to a low-resource target language, where many-shot ICL can complement the existing knowledge within the LLM. We use the target languages with the largest gap reported between LLMs and state-of-the-art systems [53], namely Bemba and Kurdish, from FLORES-200 benchmark [45]. We modify the default 1-shot MT prompt from Gemini Team [15] to include multiple translation pairs as shots from the FLORES dev split (containing 997 examples). We evaluate performance on the first 150 sentences from the test set using chrF2++ [50], a standard metric based on character and word $n$-gram overlap between generated and reference translation.

See Figure 3 for results. Similar to Robinson et al. [53], we observed small gains in the few-shot regime from 1-shot to 10-shot, particularly on Kurdish. However, when using the entire dev set for many-shot ICL, we observe improvements of 15.3% on Bemba and 4.5% on Kurdish, relative to the 1-shot Gemini prompt. Overall, these results establish the new-state-of-art for these language pairs.

---

[1]This corresponds to original version in the Gemini 1.5 Tech Report, released in February 2024. We note that the Gemini 1.5 Pro API now serves a newer version starting from April 2024.

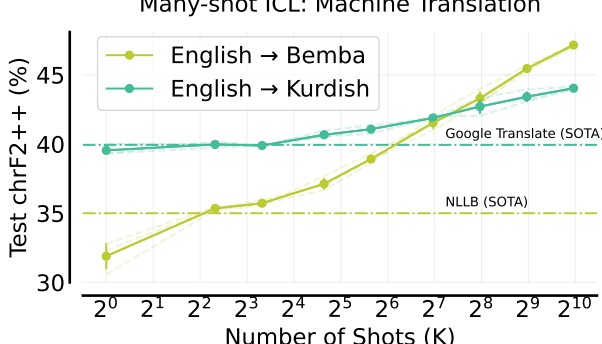

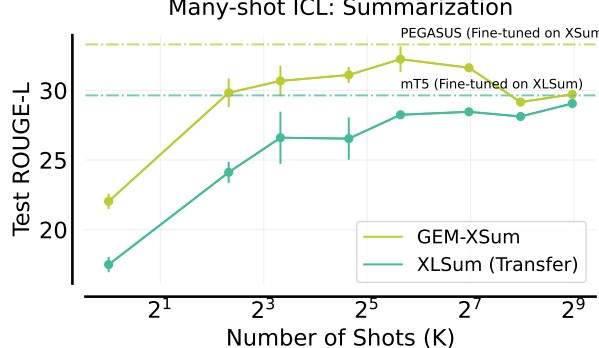

Figure 3 | **Machine Translation** (MT). Test Performance improves monotonically as we increase the number of MT pairs provided as in-context examples during inference. Notably, many-shot ICL outperforms state-of-the-art chRF2++ scores of 35% (NLLB) on Bemba and 40% (Google Translate) on Kurdish [53]. We note that 997-shot prompt corresponds to around 85K tokens. See an example prompt in Figure A.1.

Figure 4 | **Summarization**. As we increase the number of shots from XSum dev set, XSum test performance improves up to 50 shots and then deteriorates. In contrast, XLSum performance typically improves with more shots from XSum. The 500-shot prompt corresponds to 205K tokens. PEGASUS [71] and mT5 [20] are specialized models fine-tuned for summarization. See an example prompt in Figure A.2.

## 2.2. Abstractive Summarization

To investigate how scaling ICL examples can impact the comprehension ability of LLMs, we now consider abstractive news summarization using XSum dataset from the GEM benchmark [1]. Using XSum dev set examples containing news articles and summaries, we also evaluate how many-shot ICL generalizes to XLSum [20]. We report performance on 150 test articles using ROUGE-L [35], which measures the longest common subsequence between reference and generated summaries.

As depicted in Figure 4, peak performance with many-shot ICL is remarkably close to specialized models fine-tuned on XSum and XLSum. However, XSum performance declines with more than 50 in-context examples. Surprisingly, we observed the many-shot prompted model occasionally generating summaries with fabricated dates and times (§A.8), despite the absence of such data in the in-context summaries. Nonetheless, performance on XLSum monotonically improves with more shots, demonstrating positive transfer from many-shot learning to a related task.

## 2.3. Planning: Logistics Domain

Recent work has highlighted shortcomings in planning abilities of LLMs [59]. To this end, we evaluate whether many-shot ICL can improve their ability to generate simple plans on the Logistics domain, a widely used benchmark. The objective in this domain is to transport packages within cities via trucks, and between cities via airplanes. We generate a set of planning problems with 2-3 cities, 1-2 packages, 1 truck and airplane per city using a formal planning language (PDDL) generator [54], resulting in 1.3K problems for learning and 600 for evaluation. To compute optimal solutions for each problem, we use the Fast-Downward planner [21].

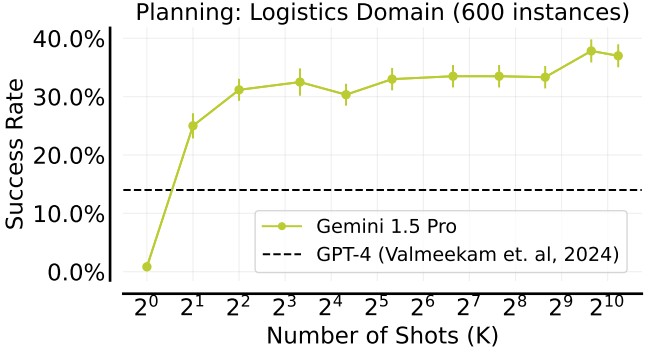

Figure 5 | **In-context Planning.** Success rate quickly improves with up to 10 shots (37K tokens), followed by saturation up to 400 shots and a sudden performance jump at 800 shots. As a reference, we report 1-shot GPT-4 results from Valmeekam et al. [59]. See Figure A.3 for an example 1-shot prompt.

As shown in Figure 5, we observe significant improvement in success rate with increasing numbers of ICL shots. While far from state-of-the-art planning approaches (e.g., Fast-Downward), our results demonstrate the potential of many-shot ICL to improve the commonsense planning abilities of LLMs.

## 2.4. Reward Modelling with Many-Shot ICL: Learning Code Verifiers

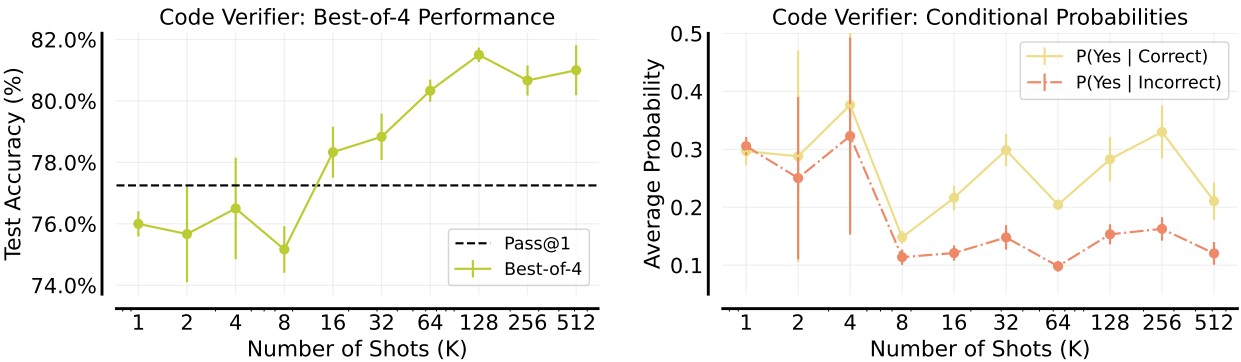

Figure 6 | **Learning Verifiers In-Context** for checking correctness of GSM8K code solutions. Error bars denotes standard error of mean over 3 seeds. See Figure A.5 for a 2-shot prompt. **Best-of-N accuracy**. (Left) Average accuracy of top-ranked code solutions (among 4 solutions) based on the verifier score on 200 GSM8K test problems. Best-of-4 selection with 128-shot bridges the gap between Pass@1 accuracy of 77.25% and Pass@4 accuracy of 90% with Gemini 1.0 Pro model. **Verifier Confidence**. (Right) Conditional Probabilities of the Yes token $\mathbb{P}(Yes)$ from the verifier, averaged over all correct and incorrect solutions on test problems.

A standard approach to improve LLM reasoning is to use test-time verification [10, 44, 24]. Specifically, an LLM generates multiple candidate solutions for a given problem and a verifier, also known as an *outcome reward* model, ranks these solutions and selects the best one. Here, we focus on learning such verifiers in-context for code verification.

To create in-context verification examples, we utilize correct and incorrect code solutions in Python generated using Gemini 1.0 Pro [15] on the GSM8K train set. In the prompt, each (problem, solution) pair is appended with the question "Is the solution correct?" followed by the Yes or No token according to ground truth correctness. At inference, we modify each test (problem, solution) pair in the same way and record the logit of the Yes and No tokens (denoted by $L_{Yes}, L_{No}$). To compute the verifier score, we use the *normalized* probability of the Yes token: $\mathbb{P}(Yes) = \exp(L_{Yes})/\big(\exp(L_{Yes}) + \exp(L_{No})\big)$. We evaluate verifier performance using best-of-4 selection based on the verifier score on 200 problems from GSM8K test set with Gemini 1.0 solutions.

As shown in Figure 6 (left), best-of-4 accuracy with the few-shot prompted verifier significantly improves above pass@1 accuracy with 16 or more in-context examples. Along with an accuracy improvement, the probabilities of the Yes token conditioned on ground-truth correct and incorrect solutions separate with increasing the number of shots up to 256, as shown in Figure 6 (right). Overall, these results show a proof-of-concept that the Gemini model becomes better at verifying correctness of solutions with many-shot ICL.

## 3. Many-shot Learning without Human-Written Rationales

Many-shot ICL could potentially be limited by the availability of high-quality human-generated rationales or demonstrations. This is particularly challenging for complex reasoning tasks, such as GPQA [52], where human-generated rationales require significant resources and expert knowledge. In this work, we explore two simple approaches for addressing this issue.

**Reinforced ICL** Recent work [55] proposed a simplified version of Reinforced Self-Training [19], demonstrating that fine-tuning using model-generated rationales can be more effective than human-generated rationales for problem-solving tasks. Inspired by their work, we introduce Reinforced ICL, where we use model-generated rationales for in-context learning. To do so, we use a zero-shot or few-shot chain-of-thought [62] prompt as a starting point to sample multiple rationales for each training problem. Then, we select rationales that obtain the correct final answer (we assume access to ground truth final answers or correctness checks), and arrange them into in-context examples containing (problem, rationale) pairs.

One potential issue with model-generated rationales is that of false positives: it is possible for an incorrect reasoning chain to lead to the correct final answer, and fine-tuning or prompting using such a reasoning chain would typically harm performance. Nevertheless, as we discuss in later sections, we often find model-generated rationales to be at least as effective human-written rationales.

**Unsupervised ICL** We now go one step further than Reinforced ICL: what if we removed rationales from the many-shot prompt altogether, and prompt the model only with inputs? Specifically, the Unsupervised ICL prompt consists of: 1) a preamble, such as, "You will be provided questions similar to the ones below:", 2) a list of unsolved inputs or problems, and 3) a zero-shot instruction or a few-shot prompt with outputs for the desired output format. See §A.2 for the exact prompts we use.

One hypothesis for how many-shot unsupervised ICL might surpass few-shot learning with human demonstrations is that, when the LLM already possesses the required knowledge to solve a task, any information inserted in the prompt that can narrow down what knowledge is needed for the task becomes helpful. This would be consistent with the view that ICL simply "locates" latent concepts (e.g., math problem-solving) the LLM acquired during pre-training [66, 22, 61]. As such, any of the prompt components – inputs, outputs, and their mapping – can help locate such concepts. While Unsupervised ICL is broadly applicable, it may not perform well, for example, when outputs are critical for specifying the task (Figure 9 and A.11).

## 3.1. Problem-solving: Hendrycks MATH & GSM8K

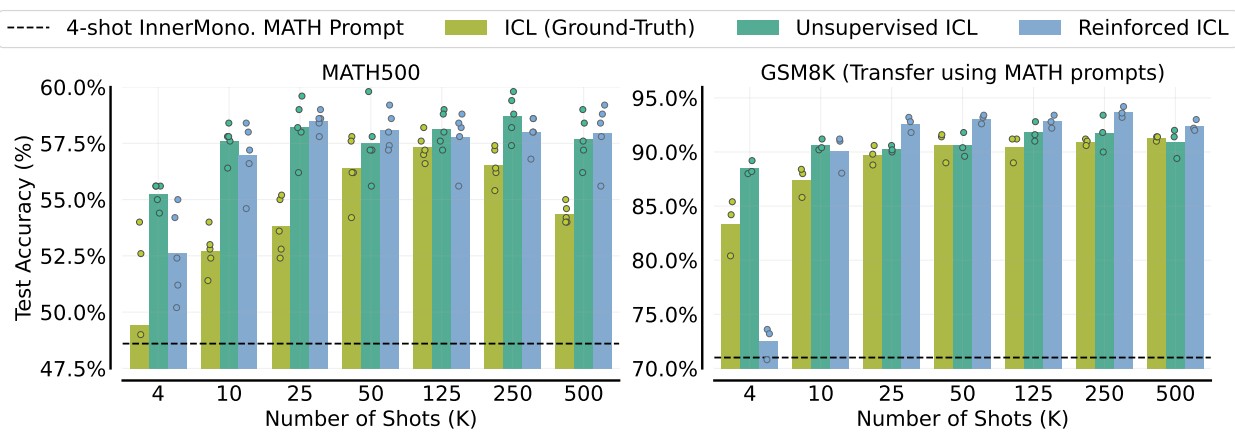

Figure 7 | **Many-shot Reinforced and Unsupervised ICL for problem-solving** generally outperform ICL with ground-truth MATH solutions. **MATH**. (Left) The bar plots depict the average performance across five random seeds on the MATH500 test set. Each random seed (denoted by the dots) corresponds to a different subset of problems along with ground truth or model-generated solutions (if any) in the prompt. **Transfer to GSM8K**. (Right) We see that the prompt obtained from MATH transfers well to the GSM8K test split containing 500 problems. Our results with many-shot ICL outperform the 4-shot Minerva prompt, which obtains a test accuracy of 55.7% on MATH500 and 90.6% on GSM8K.

We evaluate Reinforced and Unsupervised ICL on Hendrycks MATH [23], which consists of challenging high school competition-level mathematics problems. We use the MATH500 test set from Lightman et al. [33] to report performance, and our 4-shot MATH prompt for data generation can be found in Figure A.6. For Unsupervised ICL, we append this 4-shot prompt after the unsolved problems (see Figure A.8). For comparison, we also evaluate ICL with human-written solutions (ground-truth) from the MATH training set, with the same problems used for many-shot prompts.

Our results are shown in the Figure 7 (left). On MATH500, both Reinforced and Unsupervised ICL outperforms ICL with ground-truth solutions in both the few-shot and many-shot regime. For ICL, we observe that the performance improves with more examples in the prompt up to a point, and then declines (with the peak being at about 125 examples). Performance for Reinforced ICL also improves with the number of examples, and reaches a plateau at around 25 examples (while being about 5% higher than ICL), and unlike ICL, we don't see a significant drop in performance even for a very large number of examples in the context. Notably, many-shot ICL achieves comparable or superior performance when using only problems compared to using problems with solutions. This suggests solutions may be redundant for eliciting problem-solving via in-context learning on this domain, potentially due to extensive math-related data seen during pretraining.

**Can many-shot ICL enable out-of-distribution generalization?** Singh et al. [55] found that fine-tuning a model on model-generated solutions from MATH resulted in improved test performance on GSM8K [10], which has a different distribution of problems than MATH. Here, we investigate whether many-shot ICL also improves transfer performance on GSM8K, indicating an improvement in general problem-solving abilities from in-context learning. Our results in Figure 7 (right) show that this is indeed the case – Reinforced ICL with MATH prompts excels on GSM8K, outperforming ICL with ground truth MATH solutions as well as Unsupervised ICL in the many-shot setting with at least 25 shots. This indicates that model-generated solutions *can* enable better generalization than just using problems or combining them with ground-truth solutions for ICL.

## 3.2. Question Answering: Google-Proof QA (GPQA)

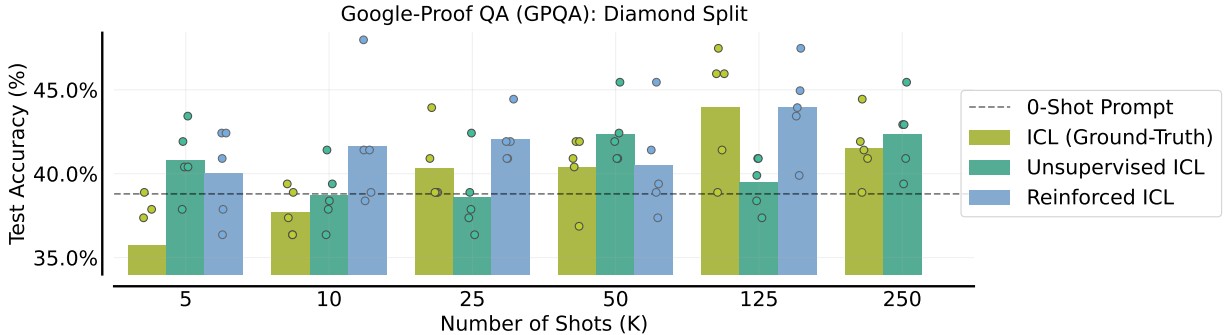

Figure 8 | **Many-shot Reinforced and Unsupervised ICL for GPQA**. The baseline zero-shot prompt, which is used for generating rationales for Reinforced ICL and appended to the prompt for Unsupervised ICL, obtains a performance of 38.8%. The average test accuracy with 125-shot prompt with both ground-truth or model-generated rationales surpass the 40.4% obtained by Claude-3 Sonnet. As we vary the number of shots, while Unsupervised ICL matches or outperforms the zero-shot prompt, Reinforced ICL consistently outperforms it.

GPQA [52] is a multiple-choice QA benchmark, with difficult questions focused on graduate-level reasoning in biology, physics, and chemistry. Following Claude-3 [3], we use the diamond split (198 problems) for evaluation. This split focuses on questions where domain experts agree but experts in other domains struggle despite extended effort and internet access. Remaining 250 questions in non-

diamond split are used for many-shot ICL with and without human-written rationales. For Reinforced ICL, we use a zero-shot prompt (Figure A.4) to generate multiple rationales on the non-diamond split, solving 129 problems. We also append this zero-shot prompt after the GPQA problems for specifying output format for Unsupervised ICL.

As shown in Figure 8, average test accuracy with ground-truth rationales improves substantially from 5 shots to 125 shots, with the best-performing 125-shot prompt nearly matching the accuracy of the state-of-the-art Claude-3 Opus. However, we do observe a performance degradation with 250 shots. Moreover, Reinforced ICL results indicate that model-generated rationales on GPQA seem to be better than ground-truth rationales up to 25 shots, while resulting in similar performance with more shots. Additionally, Unsupervised ICL does not follow any systematic trend: it sometimes performs better ICL with ground-truth rationales depending on the number of shots, but generally underperforms Reinforced ICL. As noted in Anthropic [3], GPQA is a small evaluation dataset and has an inherent higher variance across different runs, which might explain the non-systematic trends.

## 3.3. Algorithmic and Symbolic Reasoning: Big-Bench Hard

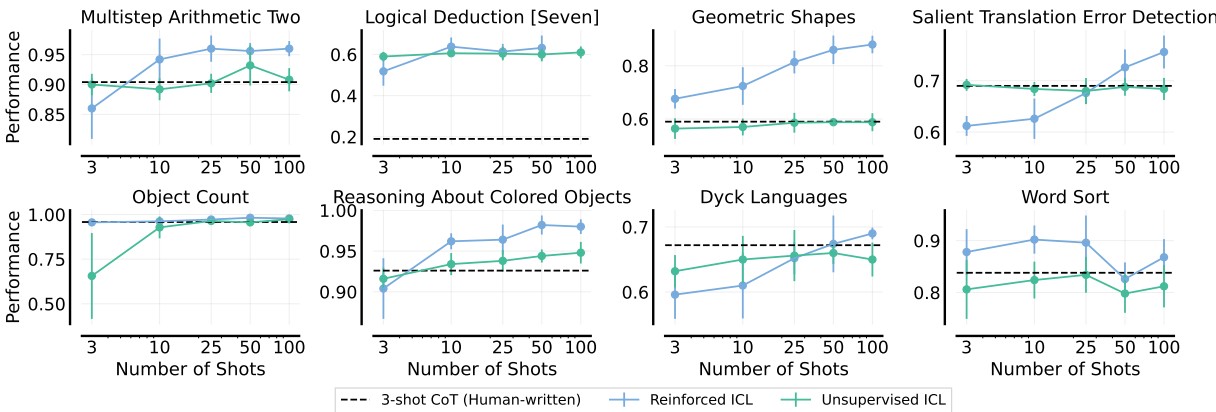

Figure 9 | **BIG-Bench Hard**. Reinforced and Unsupervised ICL with varying number of shots, averaged across five random seeds. We evaluate test performance on a held-out set of 100 problems. The error bars denote standard deviation. Reinforced ICL outperforms Unsupervised ICL for all tasks, which in turns outperforms the human-written chain-of-thought (CoT) prompt. Averaged across tasks, CoT prompting using human-written rationales gets a success rate of 72.1%, Unsupervised ICL obtains **77.1%**, while Reinforced ICL gets **83%**.

We now evaluate Reinforced ICL and Unsupervised ICL on BIG-Bench Hard [56], a suite of challenging algorithmic reasoning tasks. To reduce the impact of false positives, we select 8 tasks out of 23 in BIG-Bench Hard for which the likelihood of getting a false positive is low: either the answer string is long, or the number of options for each question is large (at least 6). For Reinforced ICL, we use the standard 3-shot CoT prompt from Suzgun et al. [56] to sample 10 rationales per problem from a training set of 150 problem at a temperature of 1.0. We filter the rationales based on final answer correctness and arrange them into prompts containing 3 to 100 (problem, rationale) pairs.

As shown in Figure 9, Reinforced ICL strongly outperforms Unsupervised ICL for almost all tasks, which in turn outperforms the standard 3-shot CoT prompt. Performance for Reinforced ICL generally improves monotonically with the number of prompts for 7 out of 8 tasks. These results indicate the Reinforced ICL is a more robust technique than Unsupervised ICL, especially for tasks in which the demonstrations contain crucial information about the task. For a few tasks, Reinforced ICL outperforms the human-written 3-shot prompt even in the 3-shot setting. This result suggests that model-generated rationales can *sometimes* outperform human-written rationales even when controlling for the amount of data, mirroring the results reported by Singh et al. [55] for fine-tuning.

# 4. Analyzing Many-Shot ICL

## 4.1. Overcoming Pre-training Biases with Many-Shot ICL

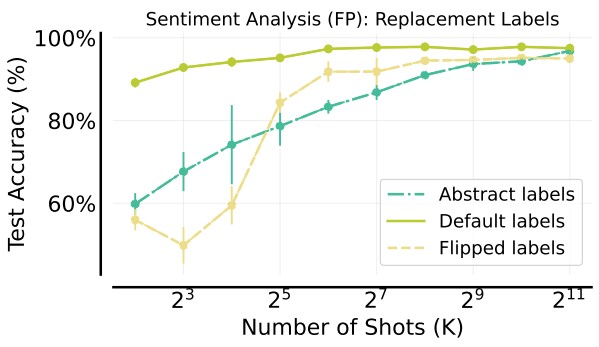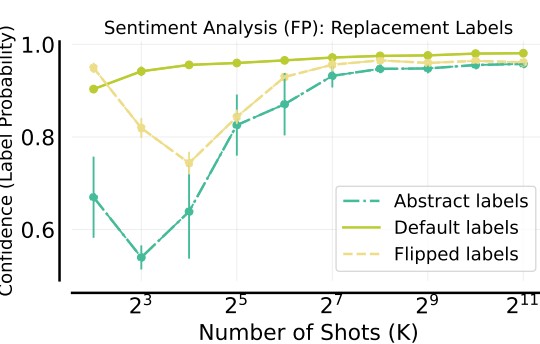

Figure 10 | **Overcoming Pre-Training Bias with Many-Shot ICL.** (Left) **Many-shot ICL overcomes label flips**: Test accuracy for sentiment analysis typically improves with more training shots. Flipped and abstract labels eventually approaching the performance of default labels. (Right) **Confidence shift in overcoming bias**. For flipped and abstract labels, model confidence in its predicted sentiment labels initially drops, then sharply increases with more training shots to similar value, suggesting a period of overcoming pre-training bias.

While LLMs demonstrate in-context learning of novel tasks, Kossen et al. [30] suggest that ICL may have difficulty unlearning biases derived from pre-training data. Their experiments, however, focused mainly on few-shot ICL due to LLM context length limitations. Here, we revisit their study using many-shot ICL on the Financial PhraseBank (FP) sentiment analysis dataset [40]. Like Kossen et al. [30], we study label relationships that affect pre-training biases:

- **Flipped Labels**: Default labels are rotated, that is, ['negative', 'neutral', 'positive'] becomes ['neutral', 'positive', 'negative']. This conflicts with sentiment biases an LLM might have learned.

- **Abstract Labels**: We use ['A', 'B', 'C'], removing any pre-existing sentiment association [63].

For ICL shots, we sample examples from the validation set (with replaced labels) to exhibit the input-label relationship and report the results in Figure 10. With few shots, test accuracy with replacement labels is much lower than with default labels. This suggests that with few-shot ICL, the model struggles to overcome its pre-existing biases from pre-training. However, as the number of shots increases, performance on flipped and abstract labels dramatically improves, approaching that of default labels. For default labels, confidence in predicted labels steadily increases with more shots, as shown in Figure 10 (right). In contrast, for flipped labels, confidence initially drops then sharply increases before reaching a plateau, suggesting a period of overcoming pre-training bias.

We posit that the initial drop in performance and confidence in the few-shot regime may be attributed to the "early ascent" phenomenon [47, 36]: a small number of shots may lead to the retrieval of an incorrect skill, which eventually diminishes as task learning takes effect in the many-shot regime. Overall, these results indicate that many-shot ICL *can* overcome pre-training biases.

## 4.2. Learning Non-Natural Language Tasks: High-Dimensional Functions

We now test many-shot ICL's ability to learn abstract mathematical functions with numerical inputs, which let us stress test its generality and applicability to possibly unseen tasks.

**Binary Linear Classification in High Dimensions**    Following the setup from Wei et al. [63], we create datasets with *N*-dimensional inputs vectors and their binary class labels, where each dimension

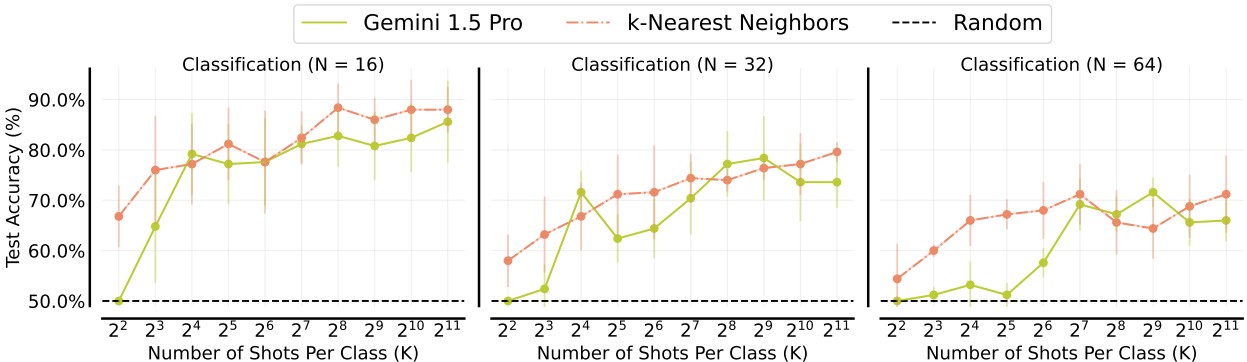

Figure 11 | **In-Context Classification**. Test accuracy for 16, 32 and 64 dimensional linear classification problems, averaged across 5 randomly-generated datasets with 25 points per class for each dataset (250 evaluation points total). As we increase the number of shots, the accuracy improves and approximately tracks the performance of the nearest-neighbor baseline trained from scratch on the same data. We use the default implementation of $k$-nearest neighbours (with $k = 5$) from scikit-learn [48]. See Figure A.7 for an example prompt.

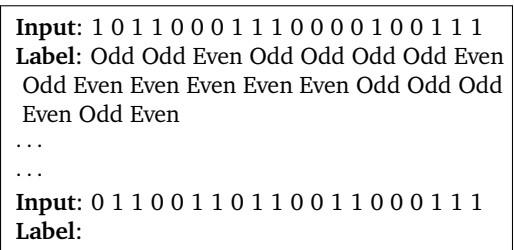

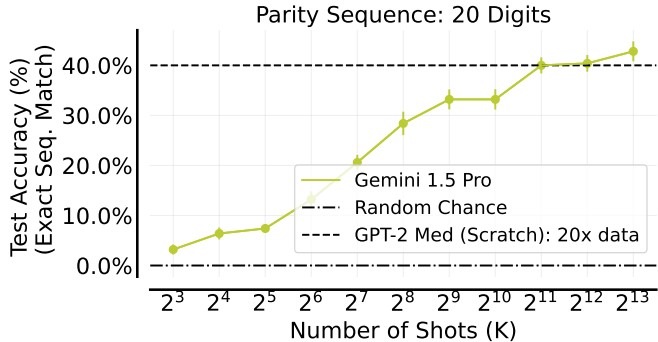

Figure 12 | **Learning Sequential Parity Function In-context**. We report test accuracy over 200 unseen inputs, averaged across 3 seeds. Error bars denote standard error of the mean. **Task Prompt**. (Left) Example prompt with input and output labels of the 20-digit Sequential Parity Function. **Test accuracy** (Right) Many-shot ICL performance improves almost monotonically with the number of shots, surpassing performance of GPT-2 Medium sized transformer trained from scratch for 1 forward-backward pass per example on 20× more data.

is a random integer in $[1, 1000]$. See more details in §A.5. While Wei et al. [63] used only 16 shots per class, we scale ICL up to 2048 shots per class. As shown in Figure 11, while 2048 shots per class perform best when $N = 16$, we observe slight accuracy decrease beyond 512 shots for higher values of $N$ (Figure 11 C, R). Moreover, many-shot ICL substantially outperforms random-chance accuracy and nearly matches the accuracy of a strong baseline, namely $k$-nearest neighbors, indicating that many-shot ICL can implement nearest-neighbour search over inputs. This is reminiscent of induction heads that implement prefix matching over sequences [46], a plausible mechanism for ICL abilities.

**Sequential Parity** Parity is a fundamental Boolean function that determines if a binary input sequence contains an even or odd number of 1s. Despite their power, transformers trained specifically for in-context learning, struggle to learn the Parity function over 20-digit sequences [7]. In this work, we evaluate how well many-shot ICL performs with a pretrained LLM to learn the sequential parity function $f(x) = [f_1(x), f_2(x), \cdots, f_n(x)]$, where $x \in \{0, 1\}^n$ and $f_i(x) = x_1 \oplus x_2 \cdots \oplus x_i \; \forall \; i \in [1, n]$. We report the results in Figure 12. We see consistent improvement in test accuracy as we increase the number of shots to 8192. Performance surpasses a GPT-2 Medium sized transformer [51] trained from scratch on 20× more input-output examples (with no repeated examples; §A.6). This result indicates many-shot ICL *can* implement computations analogous to gradient descent [60].

### 4.3. Is Many-Shot ICL Sensitive to Example Ordering?

In few-shot in-context learning (ICL), the order of examples within the prompt can significantly impact model performance [38, 65]. Here, we investigate whether such sensitivity to prompt ordering observed in few-shot ICL persists in many-shot scenarios, which remains largely unexplored. Specifically, we evaluate ten different random orderings of fixed 50 in-context examples from MATH training split and evaluate performance on the held-out MATH500 test set [33].

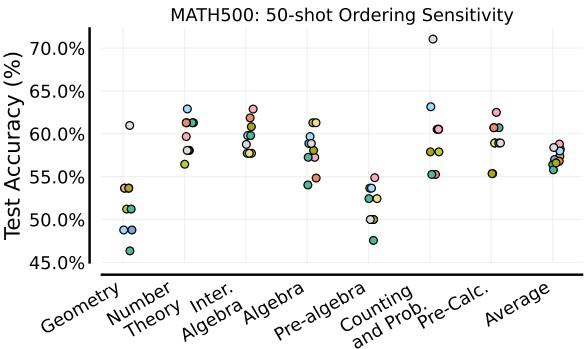

As Figure 13 reveals, performance varies significantly across different subareas in MATH500. Strikingly, an ordering that that excels in one subarea may perform poorly in another, for example, the best Geometry ordering yields weak results on Number Theory. This fluctuation results in a smaller variation in average performance compared to individual subareas. One interesting extension would be to optimize many-

Figure 13 | **Many-Shot Sensitivity To Example Ordering**. Each colored data point represents a different random ordering of 50 in-context examples provided to Gemini 1.5 Pro.

shot prompts using frameworks like DSPy [28] that has been successfully applied for optimizing few-shot prompts based a given metric. Overall, these findings highlight a key challenge in ensuring reliable results with many-shot ICL for long-context models.

### 4.4. Many-Shot ICL *vs.* Supervised Fine-Tuning

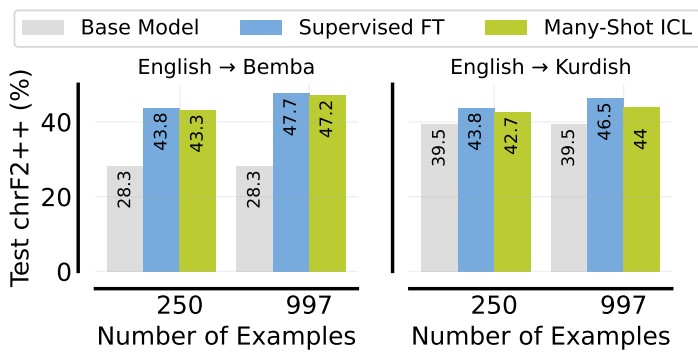

Figure 14 | **Comparing SFT with Many-Shot ICL** on low-resource translation. We plot mean performance across 3 seeds. The standard deviation is between 0.1% to 0.5%. Base model corresponds to 1-shot performance of Gemini 1.5 Pro.

Many-shot ICL could make task-specific fine-tuning less essential or, in some cases, even unnecessary, allowing LLMs to tackle a wider range of tasks without specialization. While supervised fine-tuning (SFT) is the dominant LLM paradigm when making use of hundreds or thousands of examples, it is computationally expensive in terms of training. In contrast, many-shot ICL does not require any training, however it has a larger inference cost, which can be substantially reduced with KV caching [49, 64], which might be available off-the-shelf with context caching [12].

Here, we compare many-shot ICL to full fine-tuning for machine translation (§2.1). We run two sets of experiments: one using 250 examples, and another using the entire dev set (997 examples). Our results in Figure 14 show that SFT and ICL performance is quite close for Bemba, while SFT has a slight edge for Kurdish. Overall, these results demonstrate that many-shot ICL can be a viable alternative to SFT for some tasks.

## 4.5. Comparing Many-Shot Abilities of Frontier LLMs

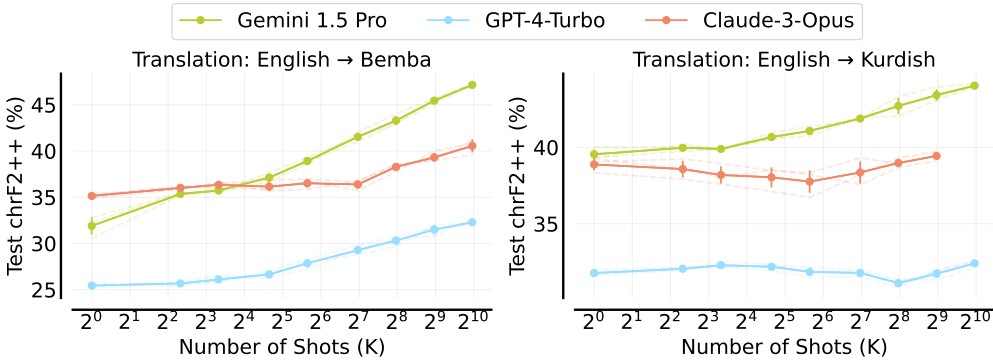

Figure 15 | Many-shot ICL with **GPT-4-Turbo** and **Claude-3-Opus** [3] on low-resource machine translation (§2.1).

The strong many-shot results with Gemini 1.5 Pro raises the question of whether other long-context frontier LLMs also benefit from many-shot ICL. To do so, we evaluate GPT-4-Turbo (128K context length) and Claude-3-Opus [3] (200K context length) on the low-resource translation (§2.1). For both these models, many-shot ICL scales favorably on Bemba but do not exhibit much improvement on Kurdish. Notably, 1.5 Pro starts lower than Claude-3 on Bemba but improves more rapidly, achieving much higher performance at 997 shots. It also outperforms GPT-4 in few-shot learning and improves further with more examples. Overall, these results indicate that frontier LLMs exhibit varying degree of many-shot ICL capability.

## 4.6. Long-context scaling laws may not predict ICL performance

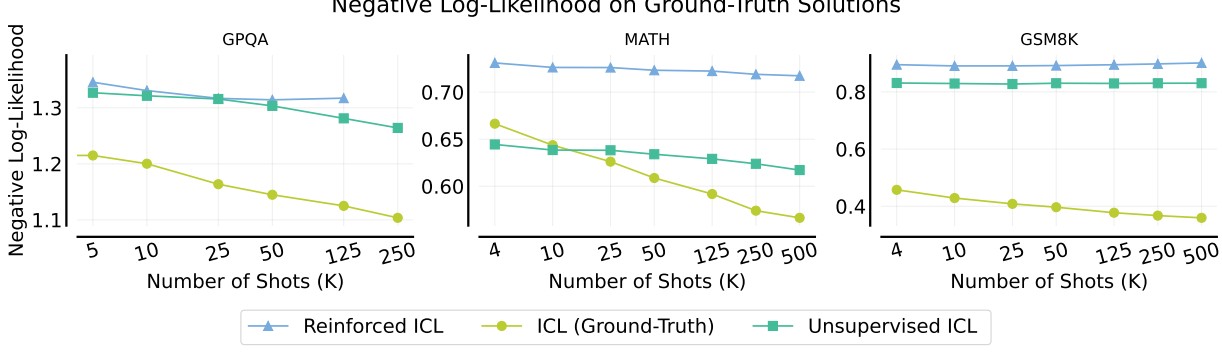

Figure 16 | **Negative Log-Likelihood** (NLL) as a function of number of shots. We plot NLL on ground truth test set solutions for GPQA, MATH and GSM8K. For GPQA and MATH, questions for Reinforced ICL and Unsupervised ICL comes from the training splits of those datasets. We study GSM8K in the transfer setting, i.e. questions for Reinforced and Unsupervised ICL come from MATH. The absolute NLL for ICL and Reinforced ICL are not directly comparable to Unsupervised ICL, since they use different prompt formats.

Prior works [68, 2, 27] have found that the negative log-likelihood (NLL) for ground-truth test outputs decreases predictably as the context length increases. We confirm this finding for GPQA, Hendrycks MATH and GSM8K with many-shot ICL, and report our results in Figure 16. However, we note that NLL trends are not a strong predictor for downstream task performance. For example, the success rate for both MATH and GPQA with ICL decreases after 125 shots (Figure 7,8), but we do not observe a corresponding increase in the NLL in Figure 16.

We also plot NLL curves for Reinforced and Unsupervised ICL, and find them to generally have a

smaller slope when compared to supervised ICL. Interestingly, NLL curves for ICL with ground-truth outputs is lower than with model-generated outputs, even though the latter often performs better. In the GSM8K transfer setting (using MATH problems and solutions to score GSM8K solutions), the change in NLL is close to nil. However, this doesn't reflect transfer performance on GSM8K, which continues to improve with more examples (Figure 7).

Overall, our results demonstrate that NLL is not a reliable proxy when attempting to predict ICL performance for problem-solving domains. This makes intuitive sense: for any given problem, there are a large number of potentially correct CoT solutions that the model can generate, and calculating the log-likelihood on only one such solution may not provide a clear picture for overall model capability. We also explore computing NLL on a diverse set of model-generated outputs on MATH, and our findings are presented in §A.7.

## 5. Related Work

**Scaling in-context learning**   Brown et al. [8] reported improved performance as you increase the number of examples (up to 64) for in-context learning in LLMs , and later works corroborated this finding [39]. However, very few works have explored using a large number of examples (1000 or above) in the prompt. This is likely due to the fact the context lengths in large language models have been quite limited until recently [16, 3]. One closely related work to ours is from Li et al. [31], who scale the number of examples for in-context learning to 2000. However, Li et al. [31] use a custom model architecture [74] to achieve long context lengths, and only evaluate models of up to 1.3B parameters, which is several orders of magnitude smaller than state-of-the-art language models, and are ineffective for complex tasks, such as GPQA [52].

Concurrently to our work, Anil et al. [2] used many-shot prompting (upto 256 shots) to jailbreak language models. In our work, we focus on a much wider range of tasks, use a lot more examples (up to 8192 shots) and use models with much longer context lengths (up to 1M tokens). Also, we explore mitigations for needing many human-generated examples with many-shot ICL. Furthermore, while Anil et al. [2] use many-shot learning to override preferences learned during RLHF phase to elicit the biases stemming from pretraining, our results in §4.1 demonstrate that we can also override pre-training biases themselves. Bertsch et al. [6] also concurrently shows benefits of scaling up in-context learning to many demonstrations on several classification datasets with up to 151 labels, albeit also using smaller context windows of up to 80k tokens (using Llama2-80k [13]).

**Long-context scaling laws**   Prior works [68, 2, 27, 16] have reported smaller next-token prediction loss with longer contexts, which Jeon et al. [25] also show using theoretical analysis. Our findings confirm this trend for even longer context lengths, but our analysis reveals some of the limitations of using next-token prediction loss as a metric for evaluating long-context performance, as next-token prediction loss continues to go down even as overall performance plateaus.

**Learning from self-generated data**   Numerous recent works [19, 70, 55] propose fine-tuning language models on self-generated data to improve performance. Their approach consists of (1) generate samples from the model and filter them using binary feedback, (2) fine-tune the model on these samples, and (3) repeat this process a few times. In this work, we extend this idea to in-context learning, and study the efficacy of Reinforced ICL in reasoning and problem-solving domains.

**Self-generated data and in-context learning**    Kim et al. [29] propose using self-generated data for few-shot ICL on classification problems, where they generate demonstrations using the LLM conditioned on the test input for each possible class label, and including these demonstrations in the context when performing the final prediction. Li et al. [32] extend this approach to reasoning and language understanding tasks, where they also generate demonstrations conditioned on the test input. Consistent with our findings, these works show that model-generated demonstrations can outperform human-generated demonstrations in the few-shot regime. Another related approach is AutoCoT [73] that uses a zero-shot CoT prompt to produce model-generated demonstrations for *few-shot* ICL. To do so, AutoCoT samples diverse questions one-by-one based on embedding-based clustering followed by heuristics-based post-processing for selecting demonstrations.

Different from above approaches, Reinforced ICL generates demonstrations using the same procedure as Singh et al. [55], does not require clustering, post-processing heuristics, or access to the test inputs for generating demonstrations, and can be applied to any problem for which we can obtain reliable reward signals. Moreover, our work mainly focuses on the utility of randomly-sampled model-generated demonstrations for many-shot ICL.

**Learning Input-Output Relationships with ICL**    Numerous works [41, 30, 69, 36] have investigated whether LLMs truly learn input-output relationships during in-context learning. Min et al. [41] found that replacing the ground truth labels in in-context examples with random labels barely effected final performance. Further investigations by Yoo et al. [69] and Kossen et al. [30] found that this finding does not necessarily hold across tasks and model sizes. In particular, Kossen et al. [30], Lin and Lee [36] showed that LLMs can indeed learn input-output relationships via in-context learning, but require more examples in order to do so well. In our work, we extrapolate the trend found in those works to much longer context lengths, showing that pre-training biases can be mostly overcome given enough training examples.

**Learning Mathematical Functions with LLMs**    Several prior works investigate whether mathematical functions can be learned with transformers [14, 72, 67, 7]. All these works train transformers specifically to perform in-context learning for such functions. In contrast, we demonstrate that many-shot ICL can learn high-dimensional functions even with pre-trained LLMs. Concurrent to our work, Vacareanu et al. [58] demonstrate that pretrained LLMs are able to perform regression tasks, with performance rivaling that of traditional supervised methods with 500 in-context examples. Our work complement their findings to other synthetic tasks with a much larger number of in-context examples. Dinh et al. [11] fine-tuned GPT-3 on synthetic classification tasks and observed similarities in the decision boundaries learned by the fine-tuned model and kNNs. Our results in Figure 11 show that many-shot ICL also performs comparably to kNNs on high-dimensional classification tasks.

**Comparing ICL with fine-tuning**    Contrary to task-specific fine-tuning, ICL does not require optimizing any model weights, allowing LLMs to perform a variety of tasks at inference. As such, several prior works compare fine-tuning with ICL but in few-shot regime. Liu et al. [37] proposed a parameter-efficient few-shot fine-tuning (FT) approach for T0 that outperforms few-shot ICL with GPT-3. However, Awadalla et al. [5] argue that few-shot ICL is more robust to distribution shifts than fine-tuning for question answering tasks. Similarly, Asai et al. [4] show better transfer with ICL compared to fine-tuning on some tasks. Mosbach et al. [42] fairly compare ICL with FT by using the same model for both approaches and show that full fine-tuning (FT) generally outperforms ICL in the few-shot regime with 16 examples. More recently, Lin et al. [34] show that few-shot ICL can outperform fine-tuning based approaches for aligning LLMs.

Complementary to prior works, we compare full fine-tuning with many-shot ICL with the same number of examples for low-resource translation. Notably, we find that many-shot ICL performs comparably to FT. Aligned with our findings, Bertsch et al. [6] concurrently show that many-shot ICL generally outperforms parameter-efficient fine-tuning (LoRA) on classification tasks. Overall, many-shot ICL and FT can exhibit comparable behaviors, which we leave for further investigation.

**Exemplar *vs.* Rule-based ICL generalization**    Chan et al. [9] indicate that ICL tends to generalize in a more exemplar-based way, compared to rule-based generalization during in-weights learning. Using a clever experiment with blocked attention, Bertsch et al. [6] also argue that the benefits of many in-context demonstrations arise from having access to more similar examples. While our results on in-context linear classification agree with this conclusion, our sequential parity results seem to contradict it. Strikingly, sequential parity was the task on which we saw the *most* improvement, whereas it should be a task that benefits *least* from seeing similar examples – after all, the nearest neighbor is always going to give the wrong answer (off by 1 bit). Chan et al. [9] do show that a transformer's inductive biases towards exemplar-based generalization can be shifted both by the training data and the model size, with larger models being less exemplar-based – perhaps this explains the contradictory findings, given that our work used a larger and much more capable model, though this remains an open question.

## 6. Discussion, Limitations and Future Work

We found significant gains in performance when going from few-shot to many-shot ICL on a wide range of tasks, including translation, summarization, planning, reward modeling, mathematical problem solving, question-answering, algorithmic reasoning, and sentiment analysis. To overcome the challenges of obtaining a large number of high-quality human-written rationales for many-shot ICL, we introduced two regimes: Reinforced ICL and Unsupervised ICL. Moreover, we demonstrate that, unlike few-shot ICL, many-shot ICL is effective at overriding pretraining biases, can learn high-dimensional functions with numerical inputs, and performs comparably to SFT.

One limitation of our work is that it mainly evaluates many-shot ICL with Gemini 1.5 Pro. That said, concurrent works [2, 6] as well as our preliminary results with GPT-4-Turbo and Claude-3-Opus (§4.5) indicate that other LLMs *can* also benefit from many-shot ICL. Future work should focus on evaluating the many-shot abilities of a wide range of long context models, as they become available. Furthermore, many-shot performance can likely serve as a valuable metric for evaluating the quality of long-context models, going beyond the prevalent needle-in-a-haystack test [26].

Another limitation of our work is that we don't completely understand why performance can sometimes degrades with more examples in the prompt (for example, for MATH). Our analysis found that negative log-likelihood trends are insufficient to explain this degradation, and future work should investigate new directions to shed light on the matter and improving many-shot ICL capabilities. Overall, we hope that this work lays a foundation for understanding and optimizing the use of long-context models for ICL, opening up a new frontier of LLM capabilities.

## Acknowledgements

We would like to thank Gheorghe Comanici for reviewing an early draft of this work. We are also grateful to Doina Precup, Aviral Kumar, Dale Schuurmans, Ankit Anand, Ross Goroshin, Urvashi Singh, Jannik Kossen, Charline Le Lan, and Daniel Toyoma for helpful discussions.

## Contribution Statement

RA initiated and led the project, ran majority of the many-shot experiments and analysis, came up with reinforced ICL, on-boarded collaborators, wrote the initial draft. AS contributed initial infra for experiments on MATH and GSM8K, ran BBH experiments, co-led the fine-tuning experiments, conducted NLL analysis on problem-solving tasks, and wrote several sections.

LZ contributed results for in-context verifier. BB contributed the planning logistics task. LR led the fine-tuning experiments. BZ contributed the many-shot results for GPT-4 and Claude-3. AA helped with GPQA, SC contributed the baseline for parity task and both helped edit the paper. AF and HL provided feedback on an early draft. HL also suggested the unsupervised ICL experiments. Others were involved in project discussions and minor edits to the paper.

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

# A. Appendix

## A.1. Example Prompts

> You are an expert translator. I am going to give you one or more example pairs of text snippets where the first is in English and the second is a translation of the first snippet into Kurdish. The sentences will be written
> English: <first sentence>
> Kurdish: <translated first sentence>
> After the example pairs, I am going to provide another sentence in English and I want you to translate it into Kurdish. Give only the translation, and no extra commentary, formatting, or chattiness. Translate the text from English to Kurdish.
>
> English: Its remnants produced showers across most of the islands, though as of yet, no damage or flooding has been reported.
> Kurdish: Li herêma Serengetîyê, Parka Neteweyî ya Serengetî ya Tanzanyayê, Cihê Parastina Ngorongoro û Cihê Parastina Gîyanewerên Nêçîrê Maswa û Cihê Parastina Neteweyî ya Masaî Mara ya Kendyayê hene.
> · · ·
> English: · · ·
> Kurdish:

Figure A.1 | Example prompt with a test input for translation from English to Kurdish on FLORES-MT benchmark in §2.1.

> I will first show a news article and then provide a very short one sentence long summary of it in fluent English.
>
> **Summarize the following article**: Burberry reported pre-tax profits of £166m for the year to March.
> A year ago it made a loss of £16.1m, hit by charges at its Spanish operations.
> In the past year it has opened 21 new stores and closed nine. It plans to open 20-30 stores this year worldwide.
> The group has also focused on promoting the Burberry brand online.
> Sales rose 7% to £1.28bn, with the company recording double-digit sales growth in Europe and Asia Pacific.
> Adjusted profit rose 23% to £215m, taking into account one-off items and a favourable exchange rate.
> Stores in London in particular benefited from favourable currency movements and increased tourism.
> "Looking forward, while mindful of the economic environment, Burberry plans to build on its strong financial position by accelerating investment in growth initiatives in retail, digital and new markets, while continuing to enhance the brand," said chief executive Angela Ahrendts.
> Burberry shares were up 7.6% at 659 pence in afternoon trading.
> **Summary**: Luxury fashion designer Burberry has returned to profit after opening new stores and spending more on online marketing

Figure A.2 | Example 1-shot prompt used for summarization on XSum and XLSum in §2.2.

**Please solve the problem**:
(define (problem logistics-c2-s1-p1-a2)
(:domain logistics-strips)
(:objects
a0 a1
c0 c1
t0 t1
l0-0 l1-0
p0
)
(:init
(AIRPLANE a0)
(AIRPLANE a1)
(CITY c0)
(CITY c1)
(TRUCK t0)
(TRUCK t1)
(LOCATION l0-0)
(in-city l0-0 c0)
(LOCATION l1-0)
(in-city l1-0 c1)
(AIRPORT l0-0)
(AIRPORT l1-0)
(OBJ p0)
(at t0 l0-0)
(at t1 l1-0)
(at p0 l1-0)
(at a0 l0-0)
(at a1 l1-0)
)
(:goal
(and
(at p0 l0-0)
)
)
)

**Your plan as plain text without formatting**:
(load-airplane p0 a1 l1-0)
(fly-airplane a1 l1-0 l0-0)
(unload-airplane p0 a1 l0-0)
done.

**Please solve the problem**:
(define (problem · · · )

**Your plan as plain text without formatting**:

Figure A.3 | An example 1-shot PDDL [17] prompt, with a test example for the Logistics domain in §2.3. Within a city, the locations are directly linked, allowing trucks to travel between any two of these locations. Similarly, cities are directly connected to each other allowing airplanes to travel between any two cities. Each city is equipped with one truck and has a designated location that functions as an airport

You will be given a multiple choice question with different choices such as (A), (B), (C), (D). Think step by step before giving a final answer to this question. Always finish your answer with 'Final Answer: (X)', where X is the correct answer choice. If none of the options match, choose the closest option as the final answer.

Figure A.4 | Zero-shot prompt for GPQA.

```
# problem:
It starts raining at 7:00 and pours heavily until its stops at 17:00 on a particular day.
On the second day, the rain takes 2 more hours than it took on the first day to stop.
On the third day, the rain pours for twice the amount of time it took on the second day.
Calculate the total time it was raining in the three days.

# solution:
def solution():
"""It starts raining at 7:00 and pours heavily until its stops at 17:00 on a particular day.
On the second day, the rain takes 2 more hours than it took on the first day to stop.
On the third day, the rain pours for twice the amount of time it took on the second day.
Calculate the total time it was raining in the three days."""
    first_day_rain_duration = 17 - 7 # 10 hours
    second_day_rain_duration = first_day_rain_duration + 2 # 12 hours
    third_day_rain_duration = second_day_rain_duration * 2 # 24 hours
    total_rain_duration = first_day_rain_duration + second_day_rain_duration + third_day_rain_duration
    result = total_rain_duration
    return result

# is the solution correct?
Yes

# problem:
Haley is getting ready to watch a comet fly over her house.
She spends two hours shopping for a telescope, half an hour getting everything set up in the backyard,
three times the setup time making snacks, and 20 minutes watching the comet.
What percentage of the total time she spent on all those activities was spent watching the comet,
rounded to the nearest percent?

# solution:
def solution():
"""Haley is getting ready to watch a comet fly over her house.
She spends two hours shopping for a telescope, half an hour getting everything set up in the backyard,
three times the setup time making snacks, and 20 minutes watching the comet.
What percentage of the total time she spent on all those activities was spent watching the comet,
rounded to the nearest percent?"""
    shopping_time = 2 # hours
    setup_time = 0.5 # hours
    snack_time = setup_time * 3 # hours
    comet_time = 20 # minutes
    total_time = shopping_time + setup_time + snack_time + comet_time
    percentage_comet_time = (comet_time / total_time) * 100
    rounded_percentage_comet_time = round(percentage_comet_time)
    result = rounded_percentage_comet_time
    return result

# is the solution correct?
No
```

Figure A.5 | An example 2-shot code correctness verifier prompt on GSM8K. Here, the solutions are generated using the Gemini 1.0 Pro [15] model. See results in §2.4.

Problem: Find the domain of the expression $\frac{\sqrt{x-2}}{\sqrt{5-x}}$.

Solution:

I need to find the domain of this expression.

This expression has two square root expressions.

I know the values under each square root must be non-negative.

So let me start by ensuring that.

I set $x - 2 \geq 0$, which simplifies to $x \geq 2$.

Then, I set $5 - x \geq 0$, which simplifies to $x \leq 5$.

But I need to make sure the denominator is not zero.

So, $5 - x$ must also be greater than zero, which gives me $x < 5$.

Therefore, the domain of the expression is $\boxed{[2, 5)}$.

Answer: $[2, 5)$.

Final Answer: The final answer is $[2, 5)$. I hope it is correct.

—

Problem: If $\det \mathbf{A} = 2$ and $\det \mathbf{B} = 12$, then find $\det(\mathbf{AB})$.

Solution:

I need to find the determinant of the product of matrices A and B.

I remember that the determinant of the product of two matrices equals the product of their determinants.

So, $\det(\mathbf{AB}) = (\det \mathbf{A})(\det \mathbf{B}) = (2)(12) = \boxed{24}$.

Answer: 24.

Final Answer: The final answer is 24. I hope it is correct.

—

Problem: Terrell usually lifts two 20-pound weights 12 times. If he uses two 15-pound weights instead, how many times must Terrell lift them in order to lift the same total weight?

Solution:

Okay, so Terrell lifts a total of $2 \cdot 12 \cdot 20 = 480$ pounds of weight with the 20-pound weights.

Well, if he switches to 15-pound weights, the total weight lifted will be $2 \cdot 15 \cdot n = 30n$ pounds, where n is the number of lifts.

I want to find the number of lifts, n, for the total weight lifted to be the same.

I equate $30n$ to 480 pounds and solve for n.

$$30n = 480$$
$$\Rightarrow \quad n = 480/30 = \boxed{16}$$

Answer: 16.

Final Answer: The final answer is 16. I hope it is correct.

—

Problem: If the system of equations

$$6x - 4y = a,$$
$$6y - 9x = b.$$

has a solution $(x, y)$ where $x$ and $y$ are both nonzero, find $\frac{a}{b}$, assuming $b$ is nonzero.

Solution:

I'm given a system of two equations.

I see that if I multiply the first equation by $-\frac{3}{2}$, I'll get another equation that has the same left-hand side as the second equation, $6y - 9x$.

Let me try that

$$6y - 9x = -\frac{3}{2}a.$$

Ah, I also know that $6y - 9x = b$, so I can equate these two equations.

So,

$$-\frac{3}{2}a = b \Rightarrow \frac{a}{b} = \boxed{-\frac{2}{3}}.$$

Answer: $-\frac{2}{3}$.

Final Answer: The final answer is $-\frac{2}{3}$. I hope it is correct.

—

Figure A.6 | 4-Shot Inner Monologue prompt used for MATH and GSM8K.

Input: 255 378 650 363 42 447 898 211 104 145 975 6 827 769 977 901
Output: Foo
Input: 111 677 874 692 540 800 771 325 295 106 980 148 275 882 246 136
Output: Foo
Input: 136 215 529 65 265 475 45 639 678 95 460 902 746 919 181 838
Output: Foo
Input: 62 583 498 50 198 277 519 22 935 351 142 369 349 272 880 125
Output: Bar
Input: 101 99 830 735 732 76 243 703 564 3 225 20 136 333 195 441
Output: Bar
Input: 242 430 80 153 39 269 898 6 530 524 89 377 238 697 212 539
Output: Bar
Input: 261 83 244 37 170 277 161 779 544 272 893 535 71 394 64 607
Output: Bar
Input: 402 863 114 193 413 905 894 143 193 288 174 646 411 938 212 285
Output: Bar
Input: 869 365 622 671 191 780 492 836 381 450 184 388 604 79 924 926
Output: Foo
Input: 548 823 66 658 380 81 779 449 641 673 94 130 258 229 299 278
Output: Bar
Input: 700 409 398 375 236 745 32 33 333 173 902 399 176 95 851 897
Output: Foo
Input: 673 211 14 221 508 752 147 309 338 23 827 980 373 861 980 946
Output: Foo
Input: 528 608 334 210 228 186 559 20 302 93 84 436 726 114 785 865
Output: Bar
Input: 117 190 66 628 31 838 183 687 598 11 187 226 381 979 171 39
Output: Bar
Input: 802 730 854 392 529 95 15 987 800 266 551 816 145 390 419 686
Output: Foo
Input: 723 701 860 30 217 633 226 477 720 839 548 880 277 178 512 585
Output: Foo
Input: · · ·
Output:

Figure A.7 | Example prompt with 8 shots per class for the linear classification in 16 dimensions, discussed in §4.2. Here, we use semantically-unrelated labels ('Foo' and 'Bar') following Wei et al. [63].

## A.2. Prompts for Unsupervised ICL

You will be provided Problems similar to the ones below:
Problem: What is the remainder when 369,963 is divided by 6?
Problem: The solution to the inequality

$$y = -x^2 + ax + b \leq 0$$

is $(-\infty, -3] \cup [5, \infty)$. Find the vertex of the parabola $y = -x^2 + ax + b$.
Problem: Let $x$ be an angle such that $\tan x = \frac{a}{b}$ and $\tan 2x = \frac{b}{a+b}$. Then the least positive value of $x$ equals $\tan^{-1} k$. Compute $k$.
Problem: Compute $\sin 0°$.
Problem: Let

$$f(x) = \begin{cases} 9x + 4 & \text{if } x \text{ is an integer,} \\ \lfloor x \rfloor + 5 & \text{if } x \text{ is not an integer.} \end{cases}$$

Find $f(\sqrt{29})$.

—

Now, I am going to give you a series of demonstrations of math Problems and Solutions. When you respond, respond only with the Solution of the final Problem, thinking step by step."

—

Problem: Find the domain of the expression $\frac{\sqrt{x-2}}{\sqrt{5-x}}$.
Solution:
I need to find the domain of this expression.
This expression has two square root expressions.
I know the values under each square root must be non-negative.
So let me start by ensuring that.
I set $x - 2 \geq 0$, which simplifies to $x \geq 2$.
Then, I set $5 - x \geq 0$, which simplifies to $x \leq 5$.
But I need to make sure the denominator is not zero.
So, $5 - x$ must also be greater than zero, which gives me $x < 5$.
Therefore, the domain of the expression is $\boxed{[2, 5)}$ .
Answer: $[2, 5)$.
Final Answer: The final answer is $[2, 5)$. I hope it is correct.

—

Problem: If $\det \mathbf{A} = 2$ and $\det \mathbf{B} = 12$, then find $\det(\mathbf{AB})$.
Solution:
I need to find the determinant of the product of matrices A and B.
I remember that the determinant of the product of two matrices equals the product of their determinants.
So, $\det(\mathbf{AB}) = (\det \mathbf{A})(\det \mathbf{B}) = (2)(12) = \boxed{24}$ .
Answer: 24.
Final Answer: The final answer is 24. I hope it is correct.

—

Problem: Evaluate $(x + y)(x - y)$ when $x = 13$ and $y = 5$.

Figure A.8 | Prompt used for Unsupervised ICL with MATH and GSM8K. We first start with a preamble saying that we are going to list a number of problems, and then we list the problems. We then give another pre-amble to specify the output format, and include up to 4 examples to fully describe this output format. As we go to the many-shot setting with hundreds of examples, we only increase the number of problems in the prompt, not the problem-solution pairs at the end.

You will be provided questions similar to the ones below:

Question:
A large gene has dozens of exons, of which the central ones code for folded triple helical repeats that connect the cytoskeleton with sarcolemma and extracellular space. Each exon usually codes for one folded triple alpha helix. The most common mutations of the gene are central exon deletions that create out-of-frame peptides and progressive degenerative organ waste. A solution is to deliver a Morpholino that recognizes the 5' end of the out-of-frame exon in pre-mRNA. The molecule prevents binding of the spliceosome and creates exon skipping and in-frame joining. Several missing exons are well tolerated by an organism. Which structure below is not involved in the proposed therapy?
(A) antisense
(B) polyA tail
(C) R-loops
(D) lariat

Question:
. . .
. . .

You will be given a multiple choice question with different choices such as (A), (B), (C), (D). Think step by step before giving a final answer to this question. Always finish your answer with 'Final Answer: (X)', where X is the correct answer choice. If none of the options match, choose the closest option as the final answer.

Figure A.9 | Unsupervised ICL Prompt for GPQA. We first start with a preamble saying that we are going to list a number of questions, and then we list the questions. We then give another preamble to specify the output format. As we go to the many-shot setting with hundreds of examples, we only increase the number of questions in the prompt.

You will be provided source sentences in English to translate in into Kurdish similar to the ones below:

English: Its remnants produced showers across most of the islands, though as of yet, no damage or flooding has been reported.
. . .
. . .

You are an expert translator. I am going to give you one or more example pairs of text snippets where the first is in English and the second is a translation of the first snippet into Kurdish. The sentences will be written
English: <first sentence>
Kurdish: <translated first sentence>
After the example pairs, I am going to provide another sentence in English and I want you to translate it into Kurdish. Give only the translation, and no extra commentary, formatting, or chattiness. Translate the text from English to Kurdish.

English: Its remnants produced showers across most of the islands, though as of yet, no damage or flooding has been reported.
Kurdish: Li herêma Serengetîyê, Parka Neteweyî ya Serengetî ya Tanzanyayê, Cihê Parastina Ngorongoro û Cihê Parastina Gîyanewerên Nêçîrê Maswa û Cihê Parastina Neteweyî ya Masaî Mara ya Kendyayê hene. English: · · ·
Kurdish:

Figure A.10 | Unsupervised ICL Prompt for the low-resource MT task. We first start with a preamble saying that we are going to list a number of source sentences, and then we list the sentences. We then give another preamble with 1 input-output example to specify the output format. As we go to the many-shot setting with hundreds of examples, we only increase the number of source sentences in the prompt.

## A.3. Unsupervised ICL on Machine Translation

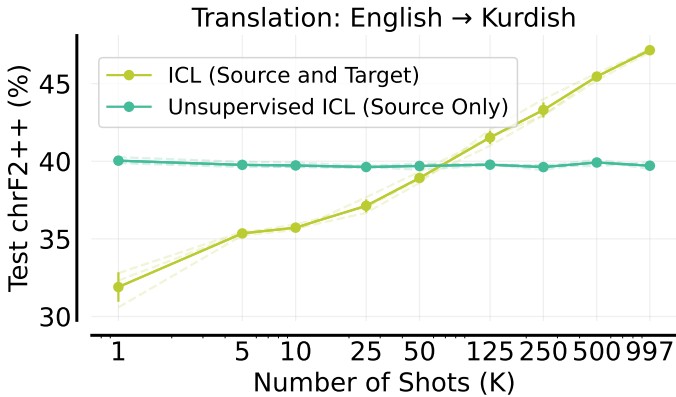

Figure A.11 | **Unsupervised ICL does not work for low-resource machine translation.** This is expected as providing only source sentences for translation task doesn't improve the task specification. See Figure A.10 for the prompt used for unsupervised ICL for this experiment.

## A.4. Reinforced ICL: Data-collection Prompt Sensitivity and Iteration 2

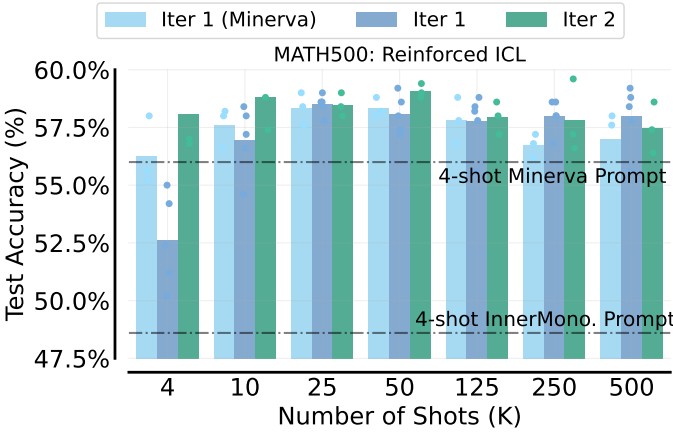

Figure A.12 | **Reinforced ICL Hendrycks MATH.** We find the performance of model-generated rationales with 4-shot Minerva prompt is generally better or comparable to the ones generated by 4-shot InnerMono. MATH prompt. Furthermore, another iteration of Reinforced ICL – generating rationales from the best performing 25-shot prompt (with model-generated rationales) on the MATH training set and using the problems which were not solved in first iteration – seem to further improve many-shot performance.

## A.5. Linear Classification: Data Generation

For each classification dataset, we randomly sample another $N$-dimensional vector as the decision boundary and a decision threshold. We then provide $K$ N-dimensional points above this threshold and $K$ points below that same threshold as in-context exemplars, and the model must determine whether unseen N-dimensional points are above or below the threshold (we do not tell the model the equation or the threshold). We provide the python code for date generation below.

```
import numpy as np

def _generate_dataset(minv, maxv, N, k, a, t):
    xtrain, ytrain = [], []
    count_pos, count_neg = 0, 0
```

```
    while (count_pos < k) or (count_neg < k):
      x_ex = np.random.randint(minv, maxv, size=N)
      label = 1
      if np.dot(x_ex, a) > t:
        if count_pos >= k:
          continue
        count_pos += 1
      else:
        if count_neg >= k:
          continue
        count_neg += 1
        label = -1
      xtrain.append(x_ex)
      ytrain.append(label)
    return np.array(xtrain).astype(str), np.array(ytrain)

def GENERATEEVAL(N, k, seed):
    """Generates one evaluation example for N-dimensional linear classification.

    Args:
        N: Dimensionality of the data.
        k: Number of in-context exemplars per class.

    Returns:
        xtrain: A list of 2k training examples (k positive, k negative).
        ytrain: A list of corresponding labels for training examples.
        xeval:  A list of evaluation examples (25 positive, 25 negative)
        yeval:  Ground-truth labels for evaluation examples.
    """

    # Step 2: Generate ground-truth coefficients
    np.random.seed(seed)
    minv, maxv = 1, 1000
    a = np.random.randint(minv, maxv, size=N)  # Random integer coefficients

    # Step 3: Generate a pivot point
    p = np.random.randint(minv, maxv, size=N)

    # Step 4: Calculate the classification threshold
    t = np.dot(a, p)

    # Steps 5: Generate training examples
    xtrain, ytrain = _generate_dataset(minv, maxv, N, k, a, t)

    # Steps 6: Generate the evaluation example
    xeval, yeval = _generate_dataset(minv, maxv, N, 25, a, t)

    return xtrain, ytrain, (xeval, yeval)
```

Listing 1 | Code for Generating Sythetic datasets for Linear Classification in High Dimensions.

## A.6. Training GPT-2 from scratch on the sequential parity task

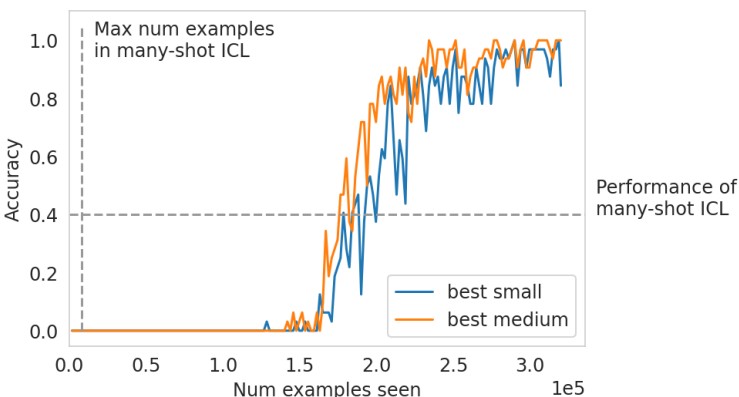

Figure A.13 | **For the sequential parity task, training a transformer from scratch does not meet 8192-shot ICL performance (dashed lines) until 20× the number of examples.** We trained two transformers on the sequential parity task (from §4.2). The smaller model was the size of GPT-2 Small, with 12 layers and 768 embedding dimension. The larger model was the size of GPT-2 Medium, with 24 layers and 1024 embedding dimension. We trained using a linear warmup and square root decay schedule, sweeping max learning rate values [1e-5, 5e-5, 1e-4, 5e-4, 1-e3] and num warmup steps [50, 100, 500, 1000, 5000]. The best values for both models (fastest learning) were max_lr=1e-4, warmup_steps=1000.

## A.7. Negative Log-Likelihood on Model-Generated Data

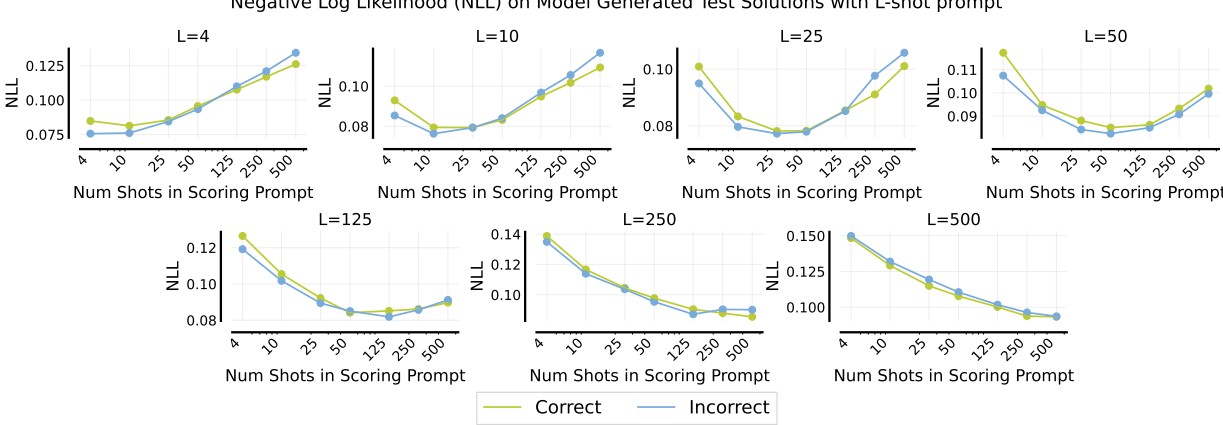

Figure A.14 | **NLL For Model-Generated Data**. We compute negative log-likelihood on model-generated solutions for Hendrycks MATH. We generate solutions using prompts of varying lengths, starting from (model-generated) 4-shot prompts, and going up to 500-shot prompts. We then score the solutions with (model-generated) prompts of different lengths, from 4-shot to 500-shot prompts. We see that, for the solutions generated using the 4-shot prompt, the NLL is lowest for 4-shot and 10-shot scoring prompts, and increases as the length of the scoring prompt increases. Note that the generation and scoring prompts contain different subsets of examples from the training sets (and are randomized for each sample). Generally speaking, we see that the NLL is lowest when the generation and the scoring prompts are of similar lengths, and higher when they are further apart. This reveals to us that prompts of similar lengths induce similar distributions over the outputs, even when the actual contents of the prompt are different. Further, we see that correct and incorrect solutions follow similar curves, and the gap between NLL for correct and incorrect solutions does not follow any specific pattern.

## A.8. Many-Shot ICL: Hallucinations on Summarization

Here is an example on XSum showing summaries with halluncinated date and time when increasing the number of shots for many-shot ICL, discussed in §2.2.

**Reference Summary**: Russia's leading independent polling agency has been labelled a "foreign agent" by the justice ministry and says it cannot now work.

**1-shot**: The Levada Centre, a Russian organization that conducts political opinion surveys, has been designated as a "foreign agent" by the government. This label restricts the organization's activities and effectively prevents it from conducting independent polls, particularly in relation to the upcoming parliamentary elections. Critics view this move as political censorship and an attempt to suppress dissent.

**5-shot**: Russia's leading independent pollster has been labelled a "foreign agent" by the justice ministry, two weeks before parliamentary elections.

**10-shot**: A leading Russian opinion pollster has been labelled a "foreign agent" by the authorities, effectively barring it from conducting any work linked to the upcoming parliamentary elections.

**25-shot**: Russia's leading independent pollster has been labelled a "foreign agent" by the justice ministry, effectively barring it from conducting election-related work.

**50-shot**: 12 September 2016 Last updated at 14:32 GMT
One of Russia's last independent pollsters has been designated a "foreign agent" by the justice ministry.

**125-shot**: 12 September 2016 Last updated at 14:43 GMT
One of Russia's last independent pollsters has been branded a "foreign agent" by the justice ministry.

**250-shot**: 2016-09-06 Last updated at 14:33 GMT
Russia's only independent pollster has been branded a "foreign agent" by the justice ministry.

**500-shot**: 12 September 2016 Last updated at 14:22 GMT
One of Russia's last independent pollsters has been branded a "foreign agent" by the government.

