# OpenReview forum: "Many-shot In-Context Learning"
_ICML.cc/2024/Workshop/ICL — ICML 2024 Workshop ICL Poster_

### Official Review · Reviewer_jcEA · 2024-06-03
**Interesting finding but improvable experiement setup**

**Rating:** 2
**Fit:** 3
**Confidence:** 3

**Workshop Review:**

The paper explores scaling the number of demonstration examples under the in-context learning settings. It’s well motivated due to longer context windows available. The findings are generally interesting, but the experiment setups are concerning. Therefore, the overall recommendation is to accept for poster.

**Clarity**:
The paper is clear and accessible overall. I’ve included some specific comments below to further improve it. The transition from Section 2 to 3 can be improved.

**Correctness**: The tasks selected are not well justified as typical LLM use-cases, and some claims are not strongly supported.

**Novelty/Interest**: This paper is a great fit for this workshop and should be interesting to most of the audience, but the technical novelty is limited.

**Specific comments (not ordered by importance)**
1. It’s good to give a formal definition of many-shot ICL in terms of the threshold where it draws the boundary from few-shot.
2. For concurrent work, I found another one on image tasks and it compares GPT-4 with Gemini as well. Link here: https://arxiv.org/abs/2405.09798
3. The idea of reinforced ICL and unsupervised ICL do not seem in parallel, so it’s better to discuss the relationship between the two to make it more organized.
4. The x-axis labels for Figure 1 should be consistent. One suggestion is to use this format: task (dataset name).
5. Figure 6, 9 and 12 are too small to read. You can consider moving some of them to appendix, but please make sure they’re legible.
6. It’s important to discuss which types of tasks are especially suitable for many-shot ICL. You hinted “difficult non-natural language tasks” in Figure 1, but it’s good to have a more comprehensive discussion.
7. It’s good to include the specific environments where the experiments run, including the model checkpoint name. The footnote at page 2 is confusing to me. Does it mean you use the February version for all the experiments? If yes, please explain why the newer version is not used.
8. What’s the implication of using KV caching? Is it critical that will affect how the results should be interpreted? If it does not affect the interpretation at all, move it to appendix rather than putting it in the main text.
10. Which version of GPT-4 used for Figure 4?
11. Can you report the performance of Fast-Downward as well? You mention ICL is worse, but by how much?
12. It might be good to apply unsupervised ICL to the first three tasks as well.
13. It’s essential to validate the overcoming of pre-training bias using more datasets (only one is used now).
14. For 4.3, did you use the same test set? If yes, why the performance (28.3% for Bemba) is different from Figure 2 (above 30% for 1-shot)?
15. It’s good that you include the specific prompts in the appendix.
16. Why ICL with ground truth is not shown in Figure 7?
17. Why the column for reinforced ICL under 250-shot is missing in Figure 6?

**Reason For Not Giving Higher Score:**

The major concern is the experiment setup (not well-justified task selection), and some claims are not strongly supported (such as the one on overcoming pre-training bias and the effectiveness of the two proposed methods based on model-generated rationale).

**Reason For Not Giving Lower Score:**

The setup is novel and the findings are in general interesting.

---

### Official Review · Reviewer_LYzm · 2024-06-03
**Review of Submission14**

**Rating:** 2
**Fit:** 3
**Confidence:** 2

**Workshop Review:**

This paper studies many-shot in-context learning, extending the few-shot paradigm to more examples. To enable many-shot learning, the paper introduces two methods: reinforced ICL, which uses model-generated rationales filtered by answer correctness, and unsupervised ICL, which removes rationales and prompts the model only with inputs. Experiments demonstrate that many-shot ICL significantly improves the performance of LLMs and overcomes pre-training biases. The paper is well-written and easy to follow. The topic is relevant to the ICL community.

**Reason For Not Giving Higher Score:**

As discussed in the Appendix, the idea of automatically generating in-context examples has been extensively studied in previous works ([1] and follow-up works). The phenomenon that increasing the number of shots improves ICL performance is not surprising. The experiment could be strengthened by discussing the effect of varying shot numbers under restricted context lengths, for example by making each example more or less informative.

[1] Automatic chain of thought prompting in large language models

**Reason For Not Giving Lower Score:**

The research topic studied in this paper is interesting and relevant to the ICL community.

---

### Meta-Review · Area_Chair_BDtr · 2024-06-14

**Recommendation:** 2

**Metareview:**

This paper investigates the setting of many-shot ICL (i.e. ICL with many examples) which has recently become possible through LLMs with large context size.

Both reviewers agree that the paper is a good fit for the workshop and they agree to recommend acceptance. Reviewer LYZM points out that generating in-context examples (one of the contributions of this paper) has been extensively studied in prior work. Reviewer jcEA points out that the technical novelty is limited and suggests to be more precise about how to define many-shot ICL.
I encourage the authors to address the detailed feedback of the reviewers.

I recommend acceptance as a poster.

---

### Decision · Program_Chairs · 2024-06-17

Accept (Poster)